# Dynamic Response of the Pitot Tube with Pressure Sensor

**DOI:** 10.3390/s23052843

**Published:** 2023-03-06

**Authors:** Jacek Pieniążek, Piotr Cieciński, Daniel Ficek, Marek Szumski

**Affiliations:** 1Department of Avionics and Control Systems, Rzeszow University of Technology, 35-959 Rzeszów, Poland; 2Department of Aerospace Engineering, Rzeszow University of Technology, 35-959 Rzeszów, Poland

**Keywords:** pressure measurement, Pitot tube, frequency response, identification, CFD modeling

## Abstract

This paper presents an attempt to determine the dynamic properties of a measuring system based on total pressure measurement with the use of a Pitot tube and a semiconductor pressure transducer. The presented research uses computed fluid dynamics (CFD) simulation and real data from the pressure measurement system for determination of the dynamical model of the Pitot tube with the transducer. An identification algorithm is applied to the data from the simulation, and the model in the form of a transfer function is an identification result. The oscillatory behavior is detected, and this result is confirmed by frequency analysis of the recorded pressure measurements. One of the resonant frequencies is the same in both experiments, but the second is slightly different. The identified dynamical models permit the possibility to predict deviations caused by dynamics and to select the appropriate tube for a particular experiment.

## 1. Introduction

Most measurements in fluid mechanics boil down to determining the velocity and/or pressure fields in a stream of fluid. Therefore, it is not surprising that many methods have been developed to accomplish this task. One of the widely used approaches is to use the coupling between the mentioned fields and limit ourselves to experimental determination of the static and total pressure fields. The technical implementation of the tests in this case boils down to the use of Pitot tubes connected to a pressure measurement device. An important problem is to determine the metrological properties of the above system in the case of variability of measured quantities. These properties can be influenced by the characteristics of the transducers subjected to external excitation, for example, generated by vibrations of the measuring system [1] or resulting from the geometry of the channel connecting the probe hole with the transducer.

In the latter case, the compressibility of the medium combined with its volume in the measurement system introduces elasticity to the measurement path and can significantly change its effectiveness in measuring a time-varying signal. This issue is the subject of this work.

An interesting analysis of the piezoresistive pressure sensor response to its rapid changes can be found in Ref. [2], but in this case, it is a surface-mounted sensor; therefore, the analysis did not consider the volume of fluid in the measuring system.

Due to the ease and low cost of implementation, as well as relatively easy interpretation of the measured value, Pitot probes occupy a prominent place among the instruments for testing pressure and flow velocity fields, which is why the literature contains a lot of information on the measurement of nonstationary flows with their use. Contemporary research problems force the reproduction of flow dynamics. The problems related to turbomachines are the best example here. Dynamic flow phenomena occur here at every step, regardless of the constancy or variability of the rotor rotational speed.

A wide range of tools and measurement methods are presented in a review article [3]. The paper also shows why pressure methods are so effective in testing turbomachines. The advantages and disadvantages of the presented techniques in comparison with thermoanemometry are analyzed here.

In the article of the A. C. Chasoglou team [4], we become acquainted with the process of designing and testing the frequency properties of a four-hole probe for studying the directional properties of turbulence. P. Duquesne [5] describes a five-hole probe focusing on the use of normalized calibration factors and their use for nonstationary quantities. It briefly presents the configuration of the calibration procedure and the measurement accuracy depending on the angle of the probe positioning. The dependence of measurement accuracy on the geometrical parameters of the probe or its setting is also described in Ref. [6]. It also includes information on measurement errors. In Ref. [7], in addition to information on the accuracy of the measurement, depending on the geometry and position of the probe, there are also data on the use and accuracy of the sensors. Similar data can also be found in Ref. [8]. A brief overview of the possibility of using several probes of different shapes and sizes is presented in article [9]. Article [10] describes the possibilities for measuring the turbulence caused by an obstacle in the form of a grid. From position [11], in turn, we can learn about the possibility of using probes for measurements at different angles of attack. The team of A. Pfau [12] presents a probe with four sensors that operate in different directions. The position [13], on the other hand, deals with the use of fast probes in transonic flows. In Ref. [14], the authors analyze and propose calibration factors for unstable flows at high temperatures and velocities approaching the speed of sound. Item [15] describes an example of measuring the total pressure field near the propeller using a simple damming probe.

Paper [16] presents preliminary results of the analysis of dynamical properties of the Pitot tube and results of the measurement conducted on the test stand of propeller propulsion. This research is a continuation and contains new results in CFD simulation (computed fluid dynamics) and new experimental data obtained using improved recording equipment. The general idea and instrumentation of measurement is presented in Section 2. Section 3 presents a short model for CFD simulation. Section 4 includes the main components of this research. Section 4.1 describes a simple estimation of the resonant frequencies. Section 4.2 presents the results of the CFD simulation and identified dynamical models of the pressure transducers with the Pitot tube. Section 4.3 presents frequency analysis of the recorded data from the measurement system. All methods give values of the resonant frequencies. Properties and usage of the identified model of dynamics are discussed in Section 5.

Papers [16,17] are the introduction to the results presented in this paper. The test stand for the research on a small propeller with a new pressure measurement system gives the possibility to measure data with a high sample rate and enhanced resolution. The main idea of this research is that the CFD simulation with pulse pressure input gives data for identification of the dynamical properties of the Pitot tube with the transducer. The frequencies response of this model is compared with the results from the propeller test stand.

## 2. Propeller Test Stand

One solution for the measurement system for dynamic pressure measurement is to integrate the transducer into the probe. The necessity of point measurement to determine the pressure field forces the miniaturization of the measuring system. One of the ways to obtain the desired dimensions is both to place a single measuring element and to make an integrated probe enabling the measurement of pressures resulting from various flow conditions. The shortening of the duct from the pressure-receiving hole to the transducer structure is an advantage here, even if the implementation technology is problematic. As shown in the previous article [16], the frequency characteristics change as a result of the presence of a pipe volume that supplies the medium to the transducer, resulting in a delay in the signal. On the other hand, the tube creates conditions for the formation of oscillations as a result of the phenomenon of oscillation.

The research concerns the analysis of the pressure field beyond the propeller. The experimental portion of the work was carried out using a stand to test small propulsion units for unmanned aerial vehicles [17]—Figure 1. In the stand, it is possible to determine the thrust of the propeller by measuring the increase in the momentum of the airstream induced by the propeller. The procedure requires knowledge of the pressure jump for air flowing through the propeller disk (between zones 1 and 3, Figure 2).

*p_s_*_0_—outer pressure,*p*_1_—pressure before propeller,*p*_2_—pressure after propeller,*p*_t_—pressure distribution after propeller.

A probe placed in zone 2 measures the variable total pressure. The observed variability is the result of the transitions of the propeller blades and the wake and helical vortex surfaces generated by them. The phenomenon of variability in the direction and speed values intensifies during the operation of the propeller in static conditions, when (because of the small pitch of the helical surfaces) a strong interaction between their vortices is observed.

The pressure measurement system used to measure the pressure in the propeller stream consists of the following parts (Figure 3):Pitot tube (A);Piezoresistive differential pressure transducer (B);Tube supplying the reference pressure from the undisturbed zone (C);Instrumental amplifier (D);Power supply for the transducer and amplifier (E).

The differential pressure (difference between the total pressure at the particular point of the flow and the reference pressure in the steady zone) is converted to a small differential voltage by transducer B. This voltage is amplified by instrumental amplifier D and recorded. During the experiment, the data acquisition system (DAQ) board controlled by a personal computer (PC) application prepared for this experiment collects the voltage samples from the instrumental amplifier and also from the detector used for RPM measurement. The samples of these two signals are presented in Figure 4. The pressure pulses and detector pulses are time-shifted due to the different angular location beyond the propeller.

The specificity of the measurement conditions and requirements demands an appropriately wide bandwidth. A simple analysis was performed. Assuming that a two-blade propeller (*NB* = 2) is under test and the rotation achieves *n* = 10,000 RPM, the basic frequency of pressure pulses is given by Equation (1). Since the purpose of the test is to determine the shape of the pulses, the measurement system bandwidth should be significantly wider.
(1)$$f=\frac{NB\cdot n}{60}=333\;\mathrm{Hz},$$

The amplifier bandwidth is about 70 kHz, and the DAQ sample rate is 200 ksps, which should be sufficient so that the characteristics of the amplifier and DAQ do not significantly disturb the pressure measurement results.

In this study, pressure duct response is investigated. The Pitot tube parameters are as follows:Outer diameter 2 mm;Inner diameter 1.3 mm;Length 17 mm.

Its simplified 2D model used in the CFD simulation is presented in Figure 5, and samples of the flow around it in Figure 6. Its geometry in 3D view is presented in Figure 7.

## 3. CFD Simulation

The need to reduce costs, as well as the constant race against time, forces researchers to use simulation techniques in the initial stage of designing the measurement system. Properly conducted CFD analysis allows the researcher to determine the initial picture of the flow, and then to carry out actual measurements for the system in the most favorable, due to expectations, flow conditions. Measurement of stagnation pressure in the vicinity of a rotating element requires consideration of the dynamic properties of the pressure probe.

To explain the phenomenon of oscillation, a simulation model was prepared using ANSYS Fluent 2021R2. The simplified model of the pressure conduit was determined by measuring the dimensions of the internal channel of the pressure sensor.

Although the actual channel has two orthogonal parts, the simulation model was simplified so that a two-dimensional axisymmetric model could be used. The main part is the volume of the Pitot tube and the pressure inlet to the sensor housing, and inside the housing of a short tube (length comparable to the diameter) is the sensor membrane.

The model used in the study, presented in Figure 5, consists of a pressure inlet (PI) and a pressure outlet (PO) [18,19,20,21]. The surface of the pressure sensor area (S) and the intake pipe (I) and test points (P1, P3, Pw) were the measurement areas and the points where the pressure and velocity were recorded. The experiment was carried out using a compressible ideal gas model of air. The purpose of modeling is to obtain data that will allow identification of the characteristics of the pressure duct. For this purpose, the following study scenario was used:Conditions airflow and pressure stabilization phase—time 0 ms;Input pressure change from initial 10 Pa to 20 Pa—time 6 ms;Input pressure change back to initial value—time 7.5 ms.

The stabilization phase is important to achieve steady flow and steady pressure conditions in the experimental environment and steady air in the pipe.

For illustration, the pressure distribution at three consecutive moments in time during phase two is presented—Figure 6. In the Appendix A the pressure change with time in the whole test volume and inside the pipe can be observed. The pressure change on the sensor surface and the pressure wave in the tube are clearly visible.

**Figure 6 sensors-23-02843-f006:**
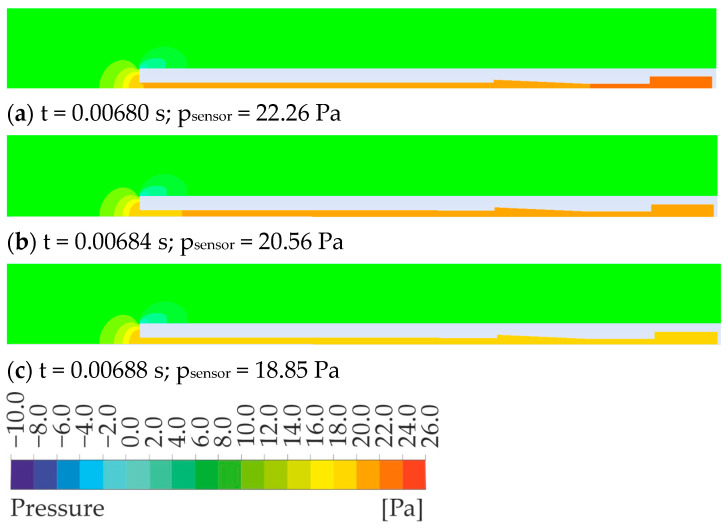
An exemplary sequence of solutions for an inlet pressure of 20 Pa.

## 4. Dynamic Model and Identification

A dynamic model of the pressure response of the tube can be identified using various methods. Simple models of pneumatic oscillators give the possibility of finding an approximate value of resonant frequency. More accurate results from CFD simulation show the fluid parameters across the volume. The appropriate experiment with pressure changes provides data on how the sensor pressure changes. The time responses of appropriate pressures being input and output are used for identification of the dynamical model. The real measurement data present pressure changes with time, but the input, i.e., real pressure in the airstream, is unknown. Although it is impossible without input to identify the model, frequency analysis shows peaks that are the results of resonant frequencies.

### 4.1. First Approximation

The entire duct from the inlet to the pressure sensitive piezoresistive bridge contains, in addition to the tube, the internal channel of the pressure transmitter. The simplified shape of this conduit is presented in Figure 5 and Figure 7.

**Figure 7 sensors-23-02843-f007:**
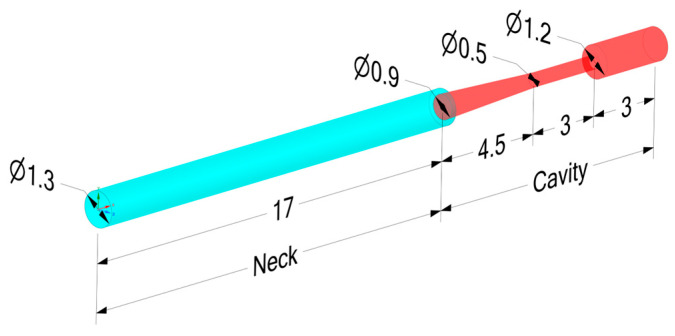
The measuring system as the Helmholtz resonator.

Some parameters of the pressure response of the pipe can be computed using simplified models. As the first approximation of the frequency characteristics of the Pitot tube with the pressure sensor, two elementary acoustic models were adopted. The first is a model of a tube with one end open, and the second is a model of the Helmholtz resonator, being a model of a tube-volume system. For the first case, the resonant frequency is described by the relationship
(2)$$f_{0}=\frac{a}{4L}\;,$$
where

*a*—speed of sound;

*L*—tube length.

For the tested geometry, it provides a frequency of about 3.2 kHz.

In the second case, the division of the volume of the measuring system was adopted as in Figure 7. For such a defined resonator, the resonant frequency should be
(3)$$f_{H}=\frac{a}{2\pi }\sqrt{\frac{A}{VL}},$$
where

*a*—speed of sound;

*A*—neck cross-section area;

*L*—neck length (in this case equal to the length of the Pitot tube);

*V*—cavity volume.

In this case, the resonant frequency was about 6.4 kHz. Of course, due to the deviations of the actual geometry from the applied model solutions, differences between the obtained results and the actual values should be expected.

### 4.2. Signal Estimation—CFD Results

The identification of the dynamics requires appropriate input which sufficiently stimulates the system [22]. A pulse of 1.5 ms in duration and 10 Pa in amplitude was used in the presented experiment. Figure 8 shows changes in the pressure value measured by the transducer, which is treated as the output signal of the measurement tube model. The pressure values at points Pw, P1 and P3 were adopted as input signals to the estimated tube model. The three pressure input signals were compared due to the expected influence of the measuring tube on the value of the observed pressure.

The data prepared for model estimation are shown in Figure 8. As can be seen, there are differences in the pressure values at the selected points. The estimation was performed using the ***n4sid*** [23] estimator from the MATLAB/Simulink package. This estimator belongs to the group of estimators using the subspace method of identification. The estimation was carried out for the three input signals pt_w, pt_1 and pt_3, and the mean value of the pressure on the sensor area p_sensor (acting directly on the transducer diaphragm) was assumed as the output signal. As a result of the above steps, three transition functions of the tube model were obtained, shown in the following transfer Functions (4)–(6).
(4)$$\frac{p\_{\mathrm{sensor}}_{}\left(z\right)}{\mathrm{pt}\_w_{}\left(z\right)}=\frac{0.0126z^{-1}-0.04115z^{-2}+0.04556z^{-3}-0.01682z^{-4}}{1-3.897z^{-1}+5.726z^{-2}-3.761z^{-3}+0.9314z^{-4}},$$
(5)$$\frac{p\_{\mathrm{sensor}}_{}\left(z\right)}{\mathrm{pt}\_1_{}\left(z\right)}=\frac{0.01115z^{-1}-0.03636z^{-2}+0.0402z^{-3}-0.01476z^{-4}}{1-3.908z^{-1}+5.761z^{-2}-3.797z^{-3}+0.9441z^{-4}},$$
(6)$$\frac{p\_{\mathrm{sensor}}_{}\left(z\right)}{\mathrm{pt}\_3_{}\left(z\right)}=\frac{0.007704z^{-1}-0.02553z^{-2}+0.02877z^{-3}-0.01075z^{-4}}{1-3.931z^{-1}+5.828z^{-2}-3.862z^{-3}+0.9651z^{-4}}$$

The sample time for these models is $T_{s}=4\;\mu s$.

In Figure 9, the original pressure response and those estimated by identified models are presented.

The accuracy of the mapping is described by fit coefficient, in which values are in the range 88.71% to 90.6% depending on the position of the reference point (Pw, P1, P3). The best representation was obtained using data from point P3, and the worst from point Pw. This is due to the fact that point Pw is close to the tube inlet, and the input and output data are more correlated than in the case of point P3, where the distance from the tube inlet is greater. Pressure at point P1 is also disturbed by the tube influence.

Frequency response of the Pitot tube obtained by the CFD simulation are shown in Figure 10. The differences in models influence the responses. The most accurate model (4) has a slightly different first resonant frequency (3.42 kHz), and the damping of both models are lower. The second eigenfrequency (6.52 kHz) is correlated with pressures at Pw and P1 that decrease identification accuracy. The resonant response of the Pitot tube influences the measured response in the case of rapid pressure change.

### 4.3. Measurement Experiment Results

The experiment using a test stand presented in Section 2 was conducted for recording pressure sensor responses in the real airstream behind a propeller. During the experiment, measurements were taken to gradually change motor RPM values, as shown in Figure 11. Figure 4 shows examples of pressure responses for two RPMs. Figure 12 shows the shape of pressure changes induced by the passage of a propeller blade.

From the entire experiment, portions of recorded data were selected for the following propeller rotational speeds:1565 RPM (f = 52.17 Hz);2970 RPM (f = 99.00 Hz);4019 RPM (f = 133.97 Hz);6750 RPM (f = 225.00 Hz).

Selected portions of data provide reference to the effects of pressure measurements in the slipstream at low and high propeller speeds. The results of frequencies analyses for selected values of rotational speed are shown in Figure 13. In Figure 13, the influence of the resonant frequencies on the pressure measurement signal is clearly visible. These are the values of 3.4 kHz and ~6.3 KHz marked in the figure.

The low frequency mean level is a result of the pressure increase due to propeller compression. The samples with different mean levels of pressure resulting from the different RPMs are presented in Figure 4 and Figure 12. The pressure fluctuations induced by propeller blades and unsteady airflow generate a signal which has noise characteristics. However, the direct periodic excitation of the frequency resulting from the propeller RPM is visible as peaks at frequencies enlisted previously in Figure 13. The peak at the first frequency of 3.4 kHz (with accuracy of about 0.1 kHz) appears in the test with 2970, 4019 and 6750 RPM. Although frequency analysis shows this resonant frequency, its amplitude is sufficiently higher than noise for 6750 RPM to be distinguishable in the time response (Figure 12). The amplitude of this mode is below noise in the experiment with 1656 RPM. This means that excitation of this mode is too small. The second frequency (6.3 kHz) is lightly detectable in experiment at 6750 RPM and does not appear for lower RPMs.

## 5. Discussion

The problem of demonstrating the influence of the Pitot tube on the measurement results is difficult to implement in practice because of the inability to measure the pressure values at the points required to determine the measurement tube model. For this reason, the authors used an indirect method, in which the basic tool for modeling of the Pitot tube is CFD simulation. The advantage of CFD simulation is the possibility of recording the pressure in the airstream flowing around the Pitot tube at any point. Numerical simulations also allow elimination of the influence of the measuring system on flow conditions that exist in real measurements.

The practical experiment was used to confirm the results obtained. In addition to that, the simple models of an open-end tube and Helmholtz oscillator provided two resonant frequencies. The resonant frequencies obtained by applying these methods are shown in Table 1.

The simple model gives a first approximation of the real parameter. The first oscillatory mode identified from the CFD results is close to the real value. The second mode’s damping and minimal excitation during the real experiment make it difficult to distinguish it among the noise. Noise may be caused by audio disturbance generated by the motor, propeller and airstream, and also by electromagnetic disturbance generated by the BLDC motor driver (similar peaks for all RPMs in Figure 13 in the high-frequency band). This disturbance covers the small components of the tube response.

Although the frequency value of the second mode is difficult to determine, it appears smaller than estimated by CFD and the Helmholtz resonator. This difference should be evaluated in further research. The first possible source of inaccuracy is simplification of the model geometry used in the 2D simulation. Three-dimensional models will be used in future research for determining the possibility of obtaining more accurate results.

The practical usage of the presented method is that for the particular geometry of pipes and pressure sensors, after CFD and identification, the model of response is obtained. This model can be used for the purpose of evaluating how dynamical properties disturb the real measurement. It is an important factor in the case of high-frequency pressure changes in the case of rotating machinery.

## Figures and Tables

**Figure 1 sensors-23-02843-f001:**
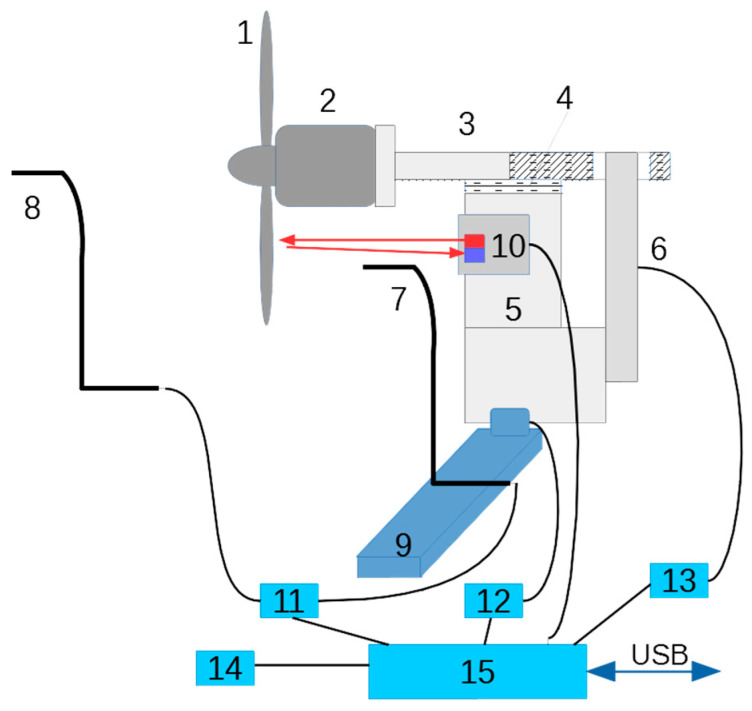
Experimental testing stand (1—propeller, 2—engine, 3—slider, 4—linear bearing, 5—base, 6—load cell, 7—Pitot tube, 8—reference Pitot tube, 9—motorized slider, 10—RPM, 11—pressure sensor, 12—stepper motor driver, 13—bridge transducer amplifier, 14—temperature sensor, 15—microcontroller system) [17].

**Figure 2 sensors-23-02843-f002:**
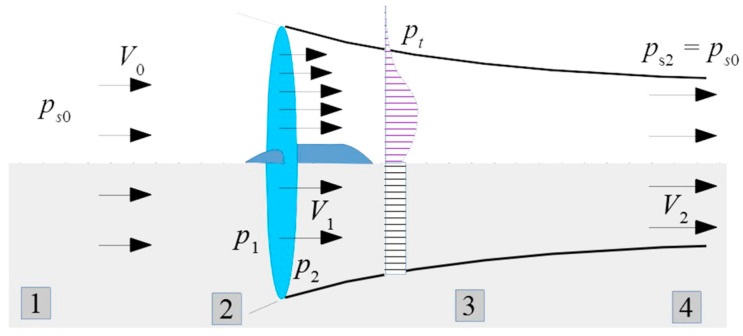
Propeller airstream: shadowed as stated by simplified actuator disk theory (1—uniform stream of incoming air or steady air, 2—unsteady area of air sucked by propeller, 3—unsteady area of air after propeller, 4—steady stream after propeller [17].

**Figure 3 sensors-23-02843-f003:**
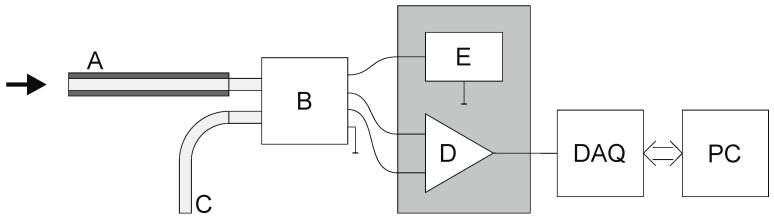
Structure of the measurement system (in-text markings) [16].

**Figure 4 sensors-23-02843-f004:**
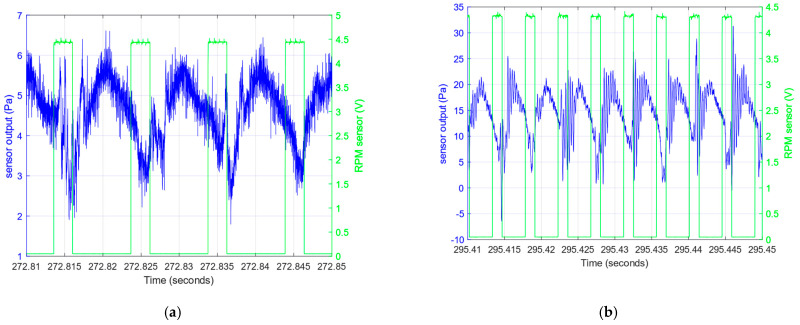
Samples of the recorded pressure responses (2970 and 6750 RPM) (blue) and pulses of the laser RPM meter (green), obtained with (**a**) 2970 RPM, (**b**) 6750 RPM.

**Figure 5 sensors-23-02843-f005:**
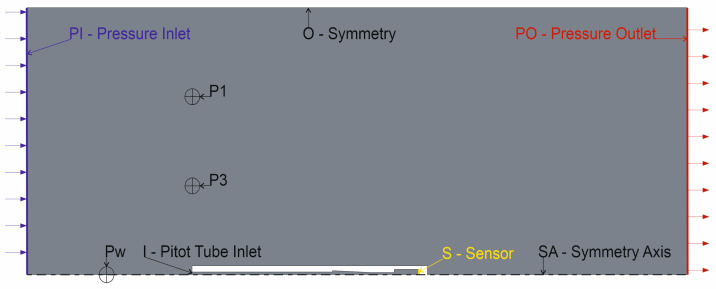
The geometry of the computational domain. The locations of the checkpoints have been marked.

**Figure 8 sensors-23-02843-f008:**
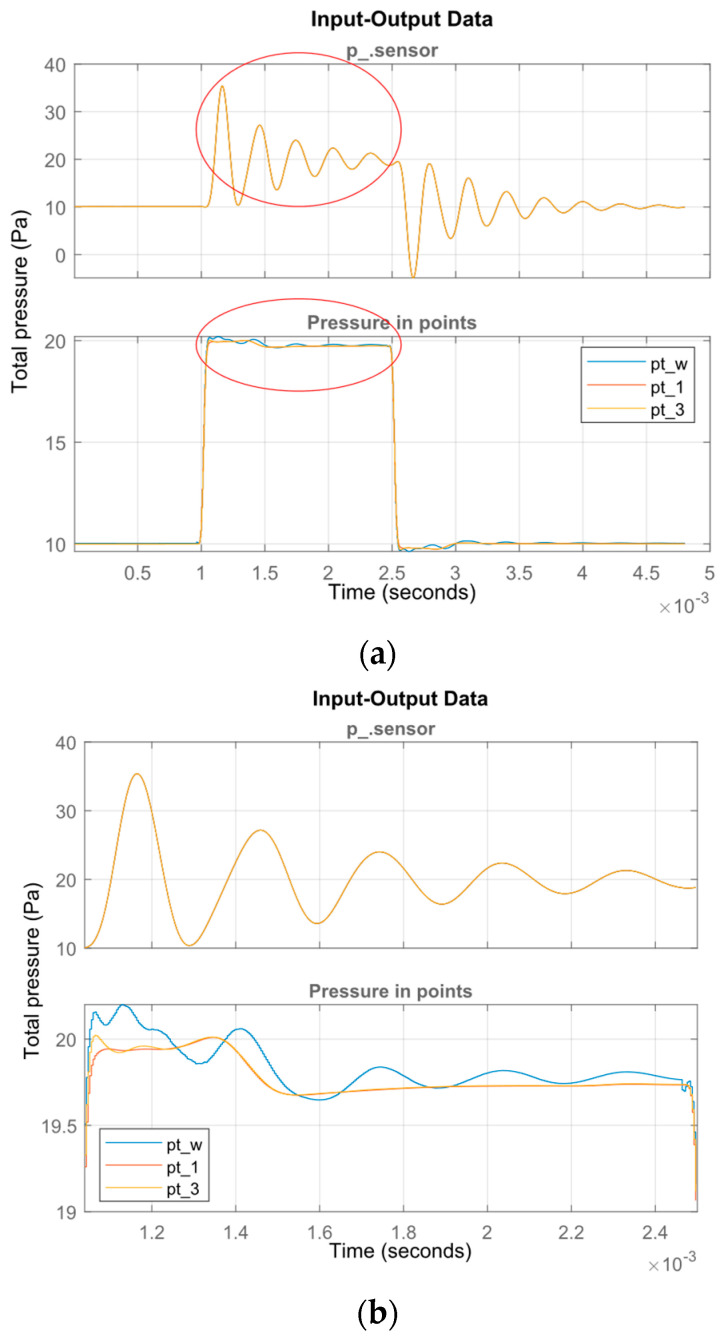
Waveforms of pressure changes at different points of the domain: full experiment (**a**), zoomed-in view of the selected portion of the chart (**b**).

**Figure 9 sensors-23-02843-f009:**
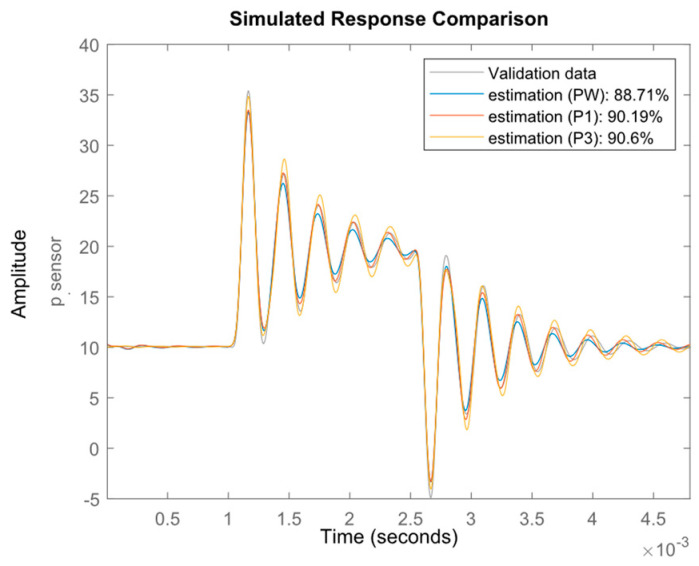
Comparison of CFD-simulated responses with estimates for different reference points.

**Figure 10 sensors-23-02843-f010:**
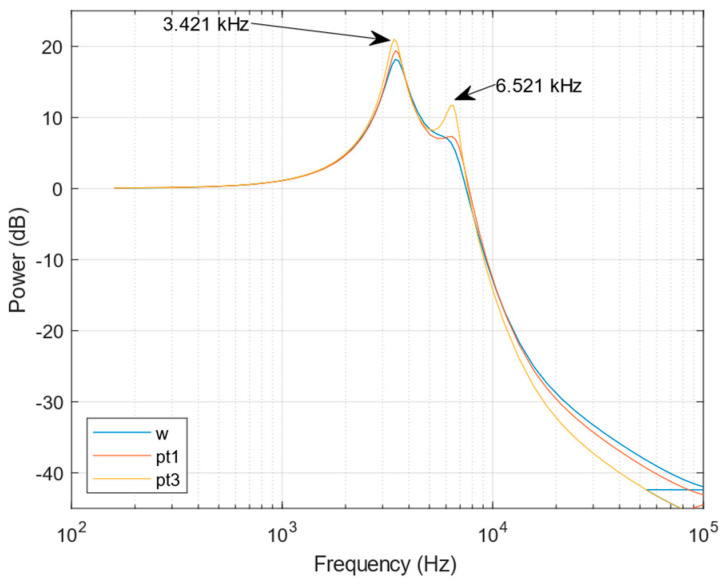
The frequency properties of the obtained Pitot tube models.

**Figure 11 sensors-23-02843-f011:**
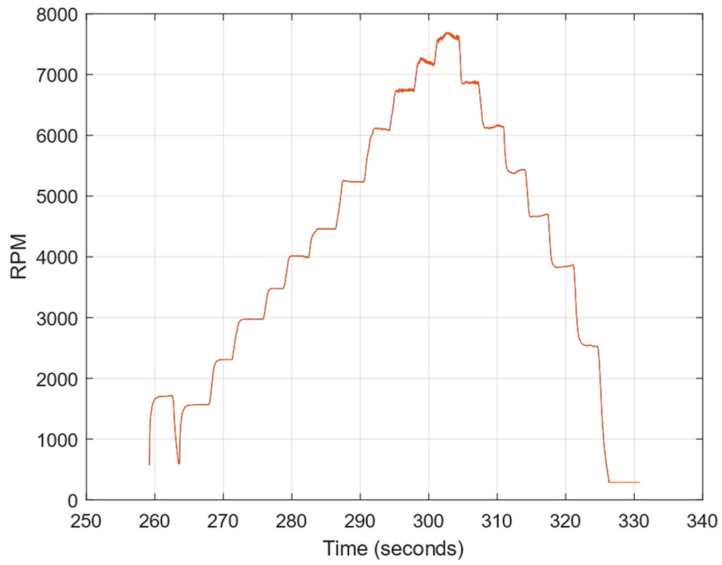
Experiment with RPM changes.

**Figure 12 sensors-23-02843-f012:**
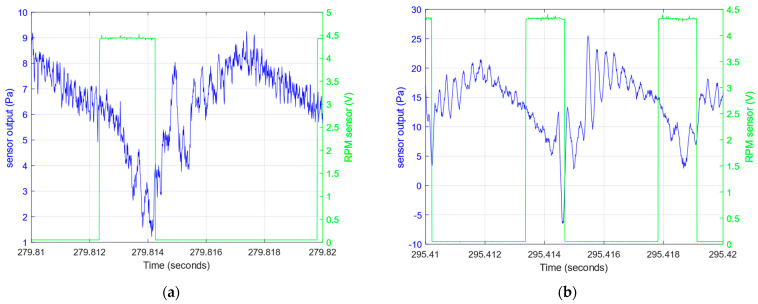
Samples of the recorded 10 ms of pressure responses (4019 (**a**) and 6750 (**b**) RPM) (blue) and pulses of the laser RPM meter (green).

**Figure 13 sensors-23-02843-f013:**
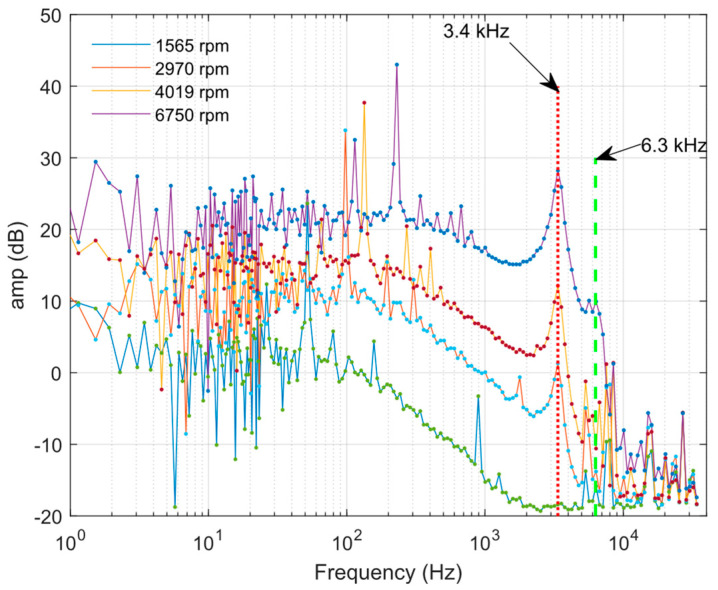
Frequency analysis results for selected RPM values, with marked frequencies.

**Table 1 sensors-23-02843-t001:** Estimated resonant frequency of Pitot tube.

Method	f_1_ (kHz)	f_2_ (kHz)
Simple acoustic models	3.2	6.4
CFD simulation	3.42	6.52
Practical experiment	3.4	~6.3

## Data Availability

Not applicable.

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
