# Peer review of "Dynamic Response of the Pitot Tube with Pressure Sensor"

_sensors, 2023, doi:10.3390/s23052843_

Round 1

Reviewer 1 Report

Dear authors.

The submitted manuscript entitled: Dynamic Response of the Total Pressure Measurement System is very interesting and addresses current challenges related to current research opportunities. However, in the submitted material there are some understatements or generalizations that make it difficult in some situations to fully understand the given results of simulations and tests.

The article introduces many symbols and notations that are not defined. I would recommend at the end of the article to put a list of abbreviations and symbols with their units and explanations, which would show whether a given parameter is simulated, calculated or measured.

There is no clear definition of where the different parameters occur. For example, line 253, 254 output signal is at the transducer diaphragm, but on Figure 12 output signal is measured by system.

There is no clear definition of where the different parameters occur. For example, line 253, 254 output signal is at the transducer diaphragm, but on Figure 12 output signal is measured by the system.

The title of the article raises the first doubt. Pressure measurements are used to determine many measured quantities, both static and dynamic. Hence, such a general title in relation to a very specific measurement, which is the measurement of pressure (indirectly, the measurement of the speed of air movement) using the Pitot tube, simply misleads the reader. I suggest making the title of the article more precise so that it reflects its content.

In Chapter 2, in which the measurement stand is described, the authors of the proposed sentence "Shortening the pipe from the pressure outlet to the transmitter structure is an advantage" - without a detailed explanation of the proposed solution, they reported that the judgment is not justified. Similarly, it is unsatisfactory to refer to costs in general terms. What are the costs and what difference in costs do the authors mean.

Other notes:

With respect to Figure 2, the authors have used symbols that are not described. Which means: pso, p1, p2, Vo, V1, pt,  ps2, V2, p1. It would also be worthwhile to accurately show the plane of rotation of the propeller.

Figure 3. What does the symbol  mean?

Figure 4. The drawing consists of two and this should be clearly defined by highlighting, e.g.: 4a)….., 4b)…..

Figure 5. What does :ww, wp mean?

Figure 7. It would be worth adding dimensions.

242        P1, P2 probably should be P1, P2.

Figure 8. The figure shows two graphs that are ambiguously described. Are "Inputs-Output Data" concern one diagram or both?. What does "wp", pt_1, pt_1, pt_3 mean.

255        Formulas:  (2)(3)(4) next new marking without description: Wp(z), Pw(z), Pt1, Pt3

Figure 12. Two graphs without a clear description. There is no indication which drawing corresponds to which rotational speed. E.g.: a)….. , b)…..

Referring to the title: "Dynamic response of the total pressure measurement system", there is no reference to testing the dynamic characteristics of the system part between the transducer diaphragm and the microcontroller system.

Small notes:

15, 216

pitot tube                    should be:       Pitot tube

249

[23]estimator              should be: [23] estimator

Taking into account the above comments and remarks, I recommend the manuscript for review and improvement.

Author Response

Point 1: The submitted manuscript entitled: Dynamic Response of the Total Pressure Measurement System is very interesting and addresses current challenges related to current research opportunities. However, in the submitted material there are some understatements or generalizations that make it difficult in some situations to fully understand the given results of simulations and tests.

The article introduces many symbols and notations that are not defined. I would recommend at the end of the article to put a list of abbreviations and symbols with their units and explanations, which would show whether a given parameter is simulated, calculated or measured.

Response 1: We put list of abbreviations and symbols at the end of the article and also correct ambiguous designations.

Point 2: There is no clear definition of where the different parameters occur. For example, line 253, 254 output signal is at the transducer diaphragm, but on Figure 12 output signal is measured by system.

Response 2: The unclearness is a result of presentation of two experiments. The measurement system used in the propeller test stand gives the voltage, which is converted to the pressure. The CFD simulation model gives the mean pressure on the sensor diaphragm area. The research methodology is clarified in the introduction. Main idea is that, the models of the Pitot tube dynamics is obtained by identification using data from CFD. The frequency responses of this model and responses from the measurement are finally compared.

Point 3: The title of the article raises the first doubt. Pressure measurements are used to determine many measured quantities, both static and dynamic. Hence, such a general title in relation to a very specific measurement, which is the measurement of pressure (indirectly, the measurement of the speed of air movement) using the Pitot tube, simply misleads the reader. I suggest making the title of the article more precise so that it reflects its content.

Response 3: According to suggestion we change the title. “Dynamic response of the Pitot tube with pressure sensor” is relevant to main subject of the article. The pressure measured by the sensor is total pressure connected to first port referenced to the pressure in the second port, which is constant static pressure or total pressure of the free stream incoming into propeller. The pressure distribution in the zone 3 (figure 2) is target measurement. The dynamic pressure and resulting flow speed are measured only in zone 4.

Point 4: In Chapter 2, in which the measurement stand is described, the authors of the proposed sentence "Shortening the pipe from the pressure outlet to the transmitter structure is an advantage" - without a detailed explanation of the proposed solution, they reported that the judgment is not justified. Similarly, it is unsatisfactory to refer to costs in general terms. What are the costs and what difference in costs do the authors mean.

Response 4: The existing pressure probes used in [2, 4, 5] are examples of the using direct pressure measurement by pressure dice. Without any analysis it is obvious that the pressure on the sensor diaphragm is real pressure at measurement point. Only disturbance is caused by the pressure probe existence what is inevitable. But design and practical realization of such kind of probe is difficult.

Point 5: With respect to Figure 2, the authors have used symbols that are not described. Which means: pso, p1, p2, Vo, V1, pt,  ps2, V2, p1. It would also be worthwhile to accurately show the plane of rotation of the propeller.

Response 5: The symbols used in Figure 2 are explained in the list of designations and in the text describing the experiment in which the described method of pressure measurement is used.

Point 6: Figure 3. What does the symbol  mean?

Response 6: Figure 3 has been corrected.

Point 7: The drawing consists of two and this should be clearly defined by highlighting, e.g.: 4a)….., 4b)…..

Response 7: The numbering has been corrected as suggested.

Point 8: Figure 5. What does :ww, wp mean?

Response 8: In this figure the sample points and areas are denoted. In the paper only three points (P1, P3, Pw) and one area (Sensor) are used so remaining ones used for the self test purposes have been removed.

Point 9: Figure 7. It would be worth adding dimensions.

Response 9: The main dimensions was in the text but now dimensions have been added in the figure.

Point 10: 242        P1, P2 probably should be P1, P2.

Response 10: Of course, it was editing error.

Point 11: Figure 8. The figure shows two graphs that are ambiguously described. Are "Inputs-Output Data" concern one diagram or both?. What does "wp", pt_1, pt_1, pt_3 mean.

Response 11: The figure was incorrectly labeled. It is corrected now.

Point 12: 255        Formulas:  (2)(3)(4) next new marking without description: Wp(z), Pw(z), Pt1, Pt3.

Response 12: The formulas (numbering corrected) are models of the dynamics in the form of the discrete transfer functions. Input and output variables names have been corrected.

Point 13: Figure 12. Two graphs without a clear description. There is no indication which drawing corresponds to which rotational speed. E.g.: a)….. , b)…..

Response 13: The numbering has been corrected as suggested.

Point 14: Referring to the title: "Dynamic response of the total pressure measurement system", there is no reference to testing the dynamic characteristics of the system part between the transducer diaphragm and the microcontroller system.

Response 14: New title clarify the subject. Today electronics makes possible to use the measurement path after the sensor fast enough to not influence the whole measurement system dynamics. The amplifier and data acquisition system has been selected considering that and the bandwidth and sample rate are far away from the characteristic frequencies of the Pitot tube.

Reviewer 2 Report

In this manuscript, the authors present the analysis of dynamic properties of Pitot tube based on CFD simulation and pressure-transducer-driven recording equipment. The work achieves the estimation of the resonant frequency of Pitot tube at customized points through theoretical acoustic models, numerical simulation, and practical experiments. The manuscript is clear and detailed. Before accepting this manuscript, there are some suggestions for modification:

1. Considering that Pitot tube has been commonly used for fluid flow measurement of the wind tunnel and flight through either experiments or simulations, the authors could highlight the novelty of this work that has not been previously demonstrated.

2. The authors mentioned the dimensions of the applied Pitot tube on Page 4. Could the influence of the parameters of the Pitot tube on the analytical results be briefly explained? 

3. What is the difference in analytical results between Pwe and Pw mentioned in Figure 5?

4. What is the unit of the y-axis in Figure 8? Is the parameter “Amplitude” dimensionless here? 

5. I suggest a zoomed-in inset be added in the second plot of Figure 8 to better illustrate the comparison of the pressure values at Pw, P1, and P3.

6. The total pressure in the Pitot tube is the sum of static pressure and dynamic pressure according to Bernoulli's equation. Could the authors briefly explain the dynamical property of their total pressure measurement system used in the manuscript?

7. There are several repeated serial numbers of the equations in the paper, for example, equation (2) and (3). Please reformat them and correctly introduce the equations in the body of the paper.

8. Some of the decimal points of the values are wrongly used with commas, such as Line 195 on Page 6, Line 226 on Page 7. Please double-check the formats in the paper and revise them.

Author Response

Point 1: Considering that Pitot tube has been commonly used for fluid flow measurement of the wind tunnel and flight through either experiments or simulations, the authors could highlight the novelty of this work that has not been previously demonstrated.

Response 1: The CFD analysis of the pressure at the sensor diaphragm is new approach to prepare dynamical model of the Pitot tube with pressure sensor. The research shows possibility to determine such response using similar to real geometry in CFD simulation. Comparison of the frequency responses from the simulation and from the dynamic pressure measurement system shows the possibility to predict how particular geometry influence response in the case of changing pressure, which is measured.

Point 2: The authors mentioned the dimensions of the applied Pitot tube on Page 4. Could the influence of the parameters of the Pitot tube on the analytical results be briefly explained?

Response 2: The simple models (part 4.1) gives the rough data of resonant frequencies, e.g. increasing length of the tube decrease f0 and fh, decreasing cross section and increasing volume increase decrease fh. The proposed approach gives possibility to find more details of the dynamic response. At this moment one configuration is considered but future research various configurations are going to be considered for the purpose to find influence of the parameters on the whole dynamics.

Point 3: What is the difference in analytical results between Pwe and Pw mentioned in Figure 5?.

Response 3: These two points (and also others) are selected to measure the dynamic pressure around the tube. Finally only points Pw, P1 and P3 are used in presented analysis. Flow conditions at Pwe becomes heavily disturbed because of the numerical phenomena near pressure inlet. In actual article version unused points have been removed from the picture.

Point 4: What is the unit of the y-axis in Figure 8? Is the parameter “Amplitude” dimensionless here?

Response 4: Amplitude is the value of pressure at the tube input. The figure labelling is corrected.

Point 5: I suggest a zoomed-in inset be added in the second plot of Figure 8 to better illustrate the comparison of the pressure values at Pw, P1, and P3.

Response 5: The zoom of the most interesting area after pressure change have been added.

Point 6: The total pressure in the Pitot tube is the sum of static pressure and dynamic pressure according to Bernoulli's equation. Could the authors briefly explain the dynamical property of their total pressure measurement system used in the manuscript?

Response 6: The target pressure measured after the propeller is a total pressure referenced to pressure in the second sensor port, so the pressure sensor respond on the difference of pressures on both propeller sides as shown on the diaphragm 1 (tubes 7 and 8).  The static pressure of free air is used as the reference pressure for the purpose to measure total pressure behind the propeller. The total pressure of the free stream incoming into  propeller is used for the purpose to measure pressure increase behind propeller. Assuming that, the reference pressure is constant, the only dynamics is a result of the pressure fluctuations inside tube and sensor.

Point 7: There are several repeated serial numbers of the equations in the paper, for example, equation (2) and (3). Please reformat them and correctly introduce the equations in the body of the paper.

Response 7: Of course, it was editing error.

Point 8: Some of the decimal points of the values are wrongly used with commas, such as Line 195 on Page 6, Line 226 on Page 7. Please double-check the formats in the paper and revise them.

Response 8: Of course, it was editing error.

Round 2

Reviewer 1 Report

Dear Authors,

Thank you for your efforts to improve the manuscript.

Therefore, I have a small note that I suggest you take into account.

It's your manuscript, so it's worth presenting it in the best possible way.

Despite my suggestions, you don't have symbol correction.

Probably, the same parameters are represented by different symbols: e.g.:

Line 174

In Figure 5, test points are marked P1, P3, PW but in the text P1, P3, Pw.

Line 269

pt_w, pt_1, pt_3,

Are the same parameters in formulas (4), (5), (6)?

Line 155, 156

In the caption you should describe what is a), b) meaning. As in Figure 8.

Line 191

There is: "time 0ms", it should be "time 0ms", 

Line 101, 102

“The construction itself, although it involves the need to determine its properties, is much less expensive than the use of available transducers.”

There is no scientific value in the above statement. Which means " less expensive" (may be cheaper). How much less? Cost may depend on calculation method etc.

Author Response

The authors would like to thank the reviewer for pointing out the shortcomings that allowed to improve the form of presentation and remove inaccuracies and obvious errors.